# Association of the Gut Microbiota with the Host’s Health through an Analysis of Biochemical Markers, Dietary Estimation, and Microbial Composition

**DOI:** 10.3390/nu14234966

**Published:** 2022-11-23

**Authors:** Maite Villaseñor-Aranguren, Carles Rosés, José Ignacio Riezu-Boj, Miguel López-Yoldi, Omar Ramos-Lopez, Anna M. Barceló, Fermín I. Milagro

**Affiliations:** 1Center for Nutrition Research, University of Navarra, 31008 Pamplona, Spain; 2Servei de Genὸmica, Universitat Autὸnoma de Barcelona, 08193 Bellaterra, Spain; 3Navarra Institute for Health Research (IdiSNA), 31008 Pamplona, Spain; 4Medicine and Psychology School, Autonomous University of Baja California, Tijuana 22390, Baja California, Mexico; 5Department of Nutrition, Food Science and Physiology, University of Navarra, 31008 Pamplona, Spain; 6Centro de Investigación Biomédica en Red de la Fisiopatología de la Obesidad y la Nutrición (CIBERobn), Carlos III Health Institute, 28029 Madrid, Spain

**Keywords:** gut microbiota, dysbiosis, metabolic health, insulin resistance, diet

## Abstract

This study aims to analyze the relationship between gut microbiota composition and health parameters through specific biochemical markers and food consumption patterns in the Spanish population. This research includes 60 Spanish adults aged 47.3 ± 11.2 years old. Biochemical and anthropometric measurements, and a self-referred dietary survey (food frequency questionnaire), were analyzed and compared with the participant´s gut microbiota composition analyzed by 16s rDNA sequencing. Several bacterial strains differed significantly with the biochemical markers analyzed, suggesting an involvement in the participant´s metabolic health. Lower levels of Lactobacillaceae and Oscillospiraceae and an increase in Pasteurellaceae, *Phascolarctobacterium*, and *Haemophilus* were observed in individuals with higher AST levels. Higher levels of the Christensenellaceae and a decrease in Peptococcaceae were associated with higher levels of HDL-c. High levels of *Phascolarctobacterium* and *Peptococcus* and low levels of *Butyricicoccus* were found in individuals with higher insulin levels. This study also identified associations between bacteria and specific food groups, such as an increase in lactic acid bacteria with the consumption of fermented dairy products or an increase in Verrucomicrobiaceae with the consumption of olive oil. In conclusion, this study reinforces the idea that specific food groups can favorably modulate gut microbiota composition and have an impact on host´s health.

## 1. Introduction

The microbiome is defined as the collective genomes of the microorganisms inhabiting a specific environment [1]. The intestinal microbiota comprises trillions of microbes and is being studied over the past years due to their physiological functions and role in the host’s health [2]. There is evidence of a bidirectional relationship between the gut microbiota and many organs in the human body, including the brain [3].

A microbial alteration is crucial to the development of pathogenesis affecting the host’s health. An alteration in the composition or function of the gut microbiota is known as dysbiosis [2], which is a word that refers to a decrease in microbial diversity, a lower amount of beneficial microorganisms, or an increase in potentially harmful microbes [4].

Several chronic disorders such as cardiovascular diseases, type 2 diabetes, inflammatory bowel disease, and non-alcoholic fatty liver disease (NAFLD) have been identified to have specific dysbiotic patterns [5]. Many of the pathologies related to gut microbiota dysbiosis are associated with metabolic complications related to insulin resistance, chronic inflammation, and oxidative stress.

The implication of gut microbiota in the host’s health is attributed to the extraction of calories from the diet, absorption of nutrients, fat deposition in adipose tissue, hepatic inflammation, and the ability to provide energy and nutrients for microbial growth and proliferation [6]. The relationship between the microbial community and the host is modulated by environmental factors such as diet and lifestyle, which may trigger changes in gut microbiota composition. The manipulation of the microbiota through nutritional strategies is considered a potential alternative when treating or preventing the development of diverse metabolic disorders.

Several patterns in the microbiota have been associated with specific dietary habits, food groups, and specific nutrients. For example, a high-fat diet with low dietary fiber and high sugars could disrupt the intestinal eubiosis and impair gut permeability. It could also contribute to the reduction in microbial diversity [7]. A diet rich in simple sugars has been associated with pro-inflammatory effects, which may mediate alterations in the epithelial integrity and impair gut microbiota composition, which can lead to the promotion of metabolic endotoxemia, systemic inflammation and metabolic dysregulation [8].

The consumption of specific foods and macronutrients, the modulation of the microbiome, and the association with health and disease are closely correlated. For example, adherence to a Mediterranean dietary pattern (MD) rich in fruit, vegetables, antioxidants, and monounsaturated and polyunsaturated fatty acids has been demonstrated to favorably modulate the gut microbiome, which probably contributes to the positive health effects of the MD [9]. These effects seemed to be caused by specific food groups characteristic of this diet.

In the present work we hypothesize that the diet, though specific compounds such as fibers, pre and probiotics, and phytonutrients, can modulate the gut microbiota. This will consequently affect the host’s health, which is evaluated by anthropometric and biochemical markers. The general aim of this study is to analyze the relationship between the participants’ gut microbiota composition and their health and nutrition status by analyzing serum biochemical markers and food intake.

## 2. Materials and Methods

### 2.1. Participants

The individuals participating in the BIOTAGUT project were recruited at the Center for Nutrition Research of the University of Navarra, Spain. The present study included 60 Spanish adults (41 females and 19 males) aged 47.3 ± 11.2 years old. The study included participants with a body mass index (BMI) between 19.0 and 34.9 kg/m^2^, without weight variations (±3 kg), changes in pharmacological treatment or consumption of stomach protectors, and gastrointestinal problems during the last three months. Subjects had to understand and be willing to sign the informed consent form and comply with all the procedures and requirements of the study. Primary exclusion criteria included a history of hypertension, cardiovascular disease, and diabetes mellitus, patients diagnosed with primary hyperlipidemia or current use of lipid-lowering drugs, and lactation or pregnancy. The study followed the ethical principles of the 2013 Helsinki Declaration [10]. The Ethical Committee in Research of the University of Navarra approved the study protocol (reference 2021.074).

### 2.2. Anthropometric and Biochemical Measurements

Anthropometric measurements, including body weight (kg) and height (cm), were collected by trained nutritionists using conventional validated methods. The SECA 216 stadiometer (SECA gmnh &co. kg, Hamburg, Germany) and the TANITA SC-330 body composition monitor (Tanita Corp, Tokyo, Japan) were used. BMI was calculated as the ratio between body weight and squared height (kg/m^2^) and was classified following the World Health Organization (WHO) standards [11].

Blood samples were drawn by venipuncture after an overnight fasting period in a clinical setting. Two tubes with EDTA and two tubes without anticoagulants were collected from each volunteer. Tubes were centrifuged for 15 min at 4500 rpm, aliquoted, and stored at −80 °C. Serum samples were used for analyzing glucose, total cholesterol (TC), high-density lipoprotein cholesterol (HDL-c), triglycerides, alanine aminotransferase (ALT), aspartate aminotransferase (AST), and insulin. These markers were analyzed with a Pentra C200 clinical chemistry analyzer (HORIBA Medical, Madrid, Spain) and suitable kits provided by the company. Serum insulin was quantified using a specific enzyme-linked immunosorbent assay (Mercodia, Uppsala, Sweden) and read with an automated analyzer system (Triturus, Grifols, Barcelona, Spain).

### 2.3. Dietary Estimation

A validated food frequency questionnaire (FFQ) that included 137 food items with corresponding portion sizes was used to estimate the habitual dietary intake [12]. Participants indicated the number of times they had consumed each food group or specific food during the previous month according to frequency categories, including daily, weekly, or monthly. Macronutrient (%) and total energy intake (kcal) were estimated with ad hoc software and the information available from valid Spanish food composition tables. Specific food groups include vegetables, fruit, legumes, cereals, whole grains, dairy intake, yogurt intake, fermented dairy products, meat, cold meat (*embutidos*), olive oil, soda, and light soda. A specific analysis of the fat consumption included total cholesterol, trans fat, monounsaturated (MUFA), polyunsaturated (PUFA), and saturated fat intake.

### 2.4. Fecal Sample Collection, DNA Extraction, and Metagenomic Data

The fecal samples were self-collected by the volunteers using OMNIgene.GUT kits from DNA Genotek (Ottawa, ON, Canada), according to the instructions provided by the company. The DNA extraction from fecal samples was performed with a QIAamp^®^ DNA kit (Qiagen, Hilden, Germany), following the manufacturer’s protocol.

Bacterial DNA sequencing was performed by the Servei de Genὸmica from the Universitat Autὸnoma de Barcelona (Bellaterra, Cerdanyola del Vallés, Spain). We analyzed the variable regions V3–V4 of the prokaryotic 16S rRNA (ribosomal Ribonucleic Acid) gene sequences, which gives 460 bp amplicons in a two-round PCR protocol. Initially, PCR is used to amplify a template out of a DNA sample using specific primers with overhang adapters attached that flank regions of interest. The full-length primer sequences were: Forward Primer: 5′TCGTCGGCAGCGTCAGATGTGTATAAGAGACAGCCTACGGGNGGCWGCAG and Reverse Primer: 5′GTCTCGTGGGCTCGGAGATGTGTATAAGAGACAGGACTACHV GGGTATCTAATCC.

PCR was performed in a thermal cycler using the following conditions: 95 °C for 3 min, 25 cycles of (95 °C for 30 s, 55 °C for 30 s, and 72 °C for 30 s), and 72 °C for 5 min. To verify that the specific primers had been correctly attached to the samples, 1 μL of the PCR product was checked on a Bioanalyzer DNA 1000 chip (Agilent Technologies, Santa Clara, CA, USA). The expected size on a Bioanalyzer was ≈550 bp.

Following this procedure, using a limited-cycle PCR, sequencing adapters, and dual indices barcodes, Nextera^®^ XT DNA Index Kit, FC-131-1002 (Illumina, San Diego, CA, USA), were added to the amplicon, which allows up to 96 libraries for sequencing on the MiSeq sequencer with the MiSeq^®^ Reagent Kit v3 (600 cycles) MS-102-3003 to be pooled together.

The libraries were quantified using a fluorometric method and dilution of the samples before pooling all samples. Finally, paired-end sequencing was performed on a MiSeq platform (Illumina) with a 600 cycles Miseq run, a 20 pM sample, and a minimum of 20% PhiX. The mean reads obtained was 164,387. Samples with more than 40,000 reads were used for further analysis. The authors deposited all the sequencing data in SRA (Sequence Read Archive), and the accession key has been included in the text (PRJNA623853).

The 16S rRNA gene sequences obtained were filtered following the quality criteria of the OTUs (operational taxonomic units) processing pipeline LotuS (release 1.58). This pipeline includes UPARSE (Highly accurate OTU sequences from microbial amplicon reads), de novo sequence clustering, and removal of chimeric sequences and phix contaminants for identifying OTUs and their abundance matrix generation. Taxonomy was assigned using HITdb (Highly scalable Relational Database). OTUs with a similarity of 97% or more were referred to as species. The abundance matrix of species, genera, families, class, order and phyla was normalized using the centered log-ratio (CLR) transformation using the R packages “compositions” and “zCompositions” [13].

### 2.5. Statistical Analysis

Microbiome Analyst [14,15] was used to analyze the participants’ microbiome composition comparing the different anthropometric measures, biochemical markers, and dietary estimation. For each variable, the population was divided into two groups (high and low levels) according to the median. To analyze the statistical differences in microbiota profiles between groups, a Zero-inflated Gaussian approach of Metagenome-Seq using the cumulative sum scaling (CSS) normalization and Student’s t-test through a CLR normalization, followed by FDR correction, were performed.

## 3. Results

### 3.1. Characteristics of the Study Population

Characteristics of the population that participated in this study, including age, anthropometric measures, and biochemical data, are shown in Table 1. Data for the whole population, and for the high and low groups according to the median, are shown.

### 3.2. Dietary Intake Characteristics

A FFQ evaluated the dietary intake characteristics of the population participating in this study. Table 2 shows the main characteristics of the dietary intake from the self-reflected questionnaire. Data for the whole population, and for the high and low groups according to the median, are shown.

### 3.3. Microbiota Composition: Biochemical Markers

The population was divided into two groups (low and high levels) according to the median of the circulating levels of the biochemical parameter, as shown in Table 1. Significant relations (FDR < 0.05) were observed between some of the biochemical markers analyzed and specific families or genera (Figure 1). No statistical differences were found at the levels of phylum, class, order and species. The biochemical markers that were significantly related to changes in the gut microbiota composition were AST, HDL cholesterol, and insulin levels. The following data are expressed as box plots in Figure 1.

Several beneficial bacteria, such as Oscillospiraceae, Lactobacillaceae, Rikenellaceae, and Porphyromonadaceae, were less abundant in the participants with higher AST levels. However, Pasteurellaceae, *Phascolarctobacterium*, and *Haemophilus* were more abundant in the same group.

Higher levels of the Christensenellaceae and lower levels of Peptococcaceae were found in the individuals with higher HDL-cholesterol. In addition, higher levels of *Phascolarctobacterium* and *Peptococcus* and a lower abundance of *Butyricicoccus* was detected in individuals with higher insulinemia. The whole summary of these results at Family and Genus levels are presented as Appendix A.

The different diversity and richness indexes (i.e., Shannon, Chao-1 or Simpson) were not associated with any of the biochemical markers in the studied population.

### 3.4. Microbiota Composition: Nutritional Markers

As shown in Table 2, the population was divided into two groups (low and high intake levels) according to the median of the intake of the different nutritional data. The bacteria whose levels were significantly different (FDR < 0.05) when comparing both groups, high and low intake of each parameter, are shown in Figure 2, Figure 3 and Figure 4. These figures represent the interface between the dietary factors presented in Table 2 and the gut bacterial composition at the different taxonomic levels. No statistical differences were found at the levels of phylum, class, order and species. 

Total energy, carbohydrate, and protein intake were associated with higher abundance of *Paraprevotella*, Enterobacteriaceae, and Unclassified *Clostridium*, respectively. Lower levels of Melainabacteriaceae were found in individuals with higher fiber intake. There were significant differences when comparing the bacterial abundance of the individuals with higher and lower consumptions of different types of fats, including monounsaturated, polyunsaturated, saturated, and trans. For example, higher levels of *Phascolarctobacterium* and *Butyricicoccus* were found in the groups with higher intake of monounsaturated and polyunsaturated fats, respectively. Higher consumption of saturated fat was associated with an increase in Acidaminococcaceae and *Phascolarctobacterium*, and a decrease in *Erysipelaclostridium.* A higher intake of trans fat was related with higher levels of *Phascolarctobacterium*.

The results in Figure 3 show higher levels of Enterobacteriaceae, Eubacteriaceae, and Streptoccaceae were observed in individuals with higher fruit consumption. Lower levels of Melainabacteriaceae were found in the group of individuals with higher consumption of whole grains. The results also show higher levels of the Lactobacillaceae family in individuals with higher consumption of yogurt and fermented dairy products. A high consumption of olive oil was related to lower amounts of *Desulfovibrio* but higher levels of *Phascolarctobacterium* and Verrucomicrobiaceae (very close taxonomically to the genus *Akkermansia*).

Individuals with higher consumption of light soda had lower abundance of the families Clostridiaceae, Methanobacteriaceae, and Dehalobacteriaceae and lower levels of the genera *Ruminococcus* and *Eggerthella*. A higher consumption of meat was related with higher levels of *Phascolarctobacterium* but a decrease in *Oscillospira*. *Butyricicoccus*, a butyrate-producing genus, was less abundant in individuals with lower consumption of cold meat (*embutidos*). 

The whole summary of the relationship between nutritional markers and microbiota composition at Family and Genus levels are presented as Appendix A, respectively. In the studied population, the different alpha-diversity indexes were not associated with any of the nutritional markers evaluated.

## 4. Discussion

### 4.1. Analysis of Microbiota with Metabolic and Hepatic Health

In the analysis between the gut microbiota and health status, we identified biochemical markers of metabolic and hepatic health that presented strong associations with specific bacteria in the microbiome. Metabolic pathologies related to obesity, insulin resistance, inflammatory conditions and metabolic endotoxemia are highly associated with dietary and lifestyle factors, which have a crucial role in the modulation of the microbiome [16]. Some specific bacteria have been related to metabolic benefits, and others have been related to a worse metabolic state [17]. Some metabolic diseases, particularly those related to insulin resistance and low-grade inflammation, have been associated with lower microbial diversity and dysbiosis. On the contrary, metabolic health and leanness have been associated with higher gut microbial diversity and richness [17].

Gut microbiota dysbiosis has been also implicated in the pathogenesis of liver diseases such as alcoholic and non-alcoholic fatty liver disease [18]. Concerning hepatic health, a decrease in Oscillospiraceae and Lactobacillaceae and an increase in Pasteurellaceae, *Phascolarctobacterium* and *Haemophilus* has been described in individuals with altered liver function through the analysis of AST and ALT [16,17,18,19,20,21,22]. Elevated levels of AST have been associated with hepatic dysfunction and higher levels of inflammation [19]. In our results, higher AST levels were related to lower abundance of *Oscillospira* and *Lactobacillus*, which are two genera that have been previously associated with health benefits in humans.

Different studies have described that metabolic pathologies are usually accompanied by gut barrier dysfunction, which may cause increased gut permeability, translocation of bacteria, and a pro-inflammatory state in the body. This state of gut dysbiosis can contribute to an increased absorption of lipopolysaccharides and metabolic endotoxemia, which has also been related to an increased risk of insulin resistance [6,17].

Our results show that some SCFA-producing bacteria, such as *Butyricicoccus,* are more abundant in participants with lower insulin resistance. This suggests that butyrate might benefit insulin metabolism. In our results, there was also an increase in the *Phascolarctobacterium* genus in individuals with higher levels of AST, as well as a significant correlation between *Phascolarctobacterium*, insulin levels and trans-fat intake. Previous evidence stated that an increase in the *Phascolarctobacterium* was associated with a decrease in bacterial diversity in patients with NAFLD, type 2 diabetes, and hepatitis B infection [23].

Critical cardiometabolic parameters and dyslipidemia (i.e., low HDL-c) have been negatively associated with higher BMI and abdominal obesity. These parameters can be modulated by dietary and lifestyle factors that contribute to changes in gut microbiota composition [7,24]. In our results, individuals with higher levels of HDL-c presented a higher abundance of Christensenellaceae, suggesting a possible relation between Christensenellaceae and lower cardiometabolic risk [25].

On the other hand, some bacteria may contribute to impair the cardiometabolic state. For example, low levels of Peptococcaceae have been associated with higher levels of HDL-c, whereas higher levels of Peptococcaceae have been positively correlated with insulin resistance [26]. Insulin resistance is associated with worse metabolic health and several cardiometabolic risk factors, such as low HDL-c [27]. Therefore, lower levels of Peptococcaceae and higher HDL-c seem to be associated with better metabolic health [28]. In our study, we have observed that Peptococcaceae was negatively associated with HDL-cholesterol.

### 4.2. Analysis of Microbiota and Dietary Intake

Diet and lifestyle are the most important modulators of gut microbiota composition. There are complex interactions between gut microbiota, dietary factors and the genetic background that are crucial for the development of metabolic syndrome features [29]. In this research, the consumption of specific food groups, such as fermented dairy products, meat, fat, and olive oil, was significantly associated with specific bacteria in the gut microbiome. For example, we found that a high caloric intake was associated with an increase in *Paraprevotella* genus and Enterobacteria. These results contribute to previous evidence stating that an elevated consumption of simple sugars and carbohydrates is associated with a worse metabolic state due to the impairment in epithelial integrity and increased inflammation [30].

The association between the consumption of fermented dairy products and an increase in *Lactobacilli* has been previously reported. Different strains from this bacterial group are used as probiotics and contribute to the prevention of obesity and other metabolic pathologies [26,31]. In this context, we have also observed that there were higher *Lactobacillus* levels in the individuals with higher consumption of fermented dairy products and yogurt.

High consumption of meat has been associated with an increase in inflammatory markers and adiposity [32]. Our results show a decrease in the levels of *Oscillospira*, a putative beneficial genus, in the individuals with higher meat consumption, suggesting a negative correlation between *Oscillospira* and meat intake.

Extra virgin olive oil has been extensively associated with anti-inflammatory effects [33]. Our results show that olive oil consumption is associated with an increase in the abundance of Verrucomicrobiaceae, which is very close taxonomically to the genus *Akkermansia*. *Akkermansia muciniphila* has been negatively associated with overweight, obesity, hypertension, and type 2 diabetes [34].

On the contrary, a high-fat diet rich in saturated and trans fats has been associated with a pro-inflammatory state characterized by a reduction in microbial diversity, increased intestinal permeability, and lipopolysaccharide translocation [35]. In our study, individuals with higher consumption of saturated and trans fat had higher amounts of *Phascolarctobacterium*, which is a lipolytic genus that secretes extracellular esterase to break down triglycerides and its hydrolyzates [36].

## 5. Conclusions

This study suggests associations between gut microbiota composition, hepatic health, and insulin resistance status, where the consumption of specific foods related with inflammation features plays an important modulatory role.

## Figures and Tables

**Figure 1 nutrients-14-04966-f001:**
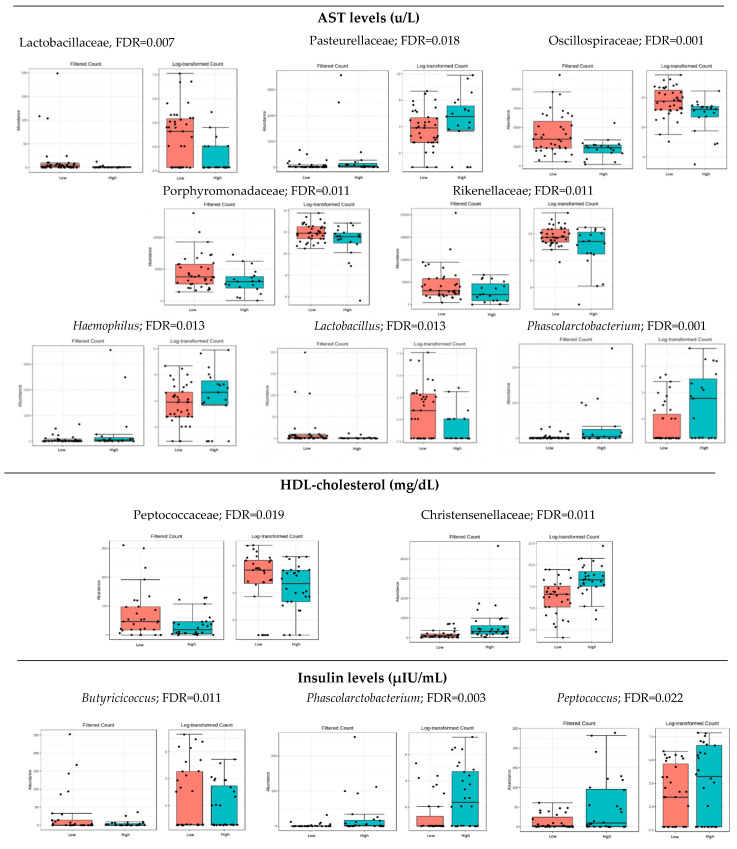
Bacterial taxa differing significantly in abundance when comparing the groups of individuals with high and low levels of biochemical biomarkers (FDR < 0.05): AST, HDL-cholesterol and insulin. Red boxes represent participants with lower levels, and blue boxes represent participants with higher levels of the specified biochemical marker.

**Figure 2 nutrients-14-04966-f002:**
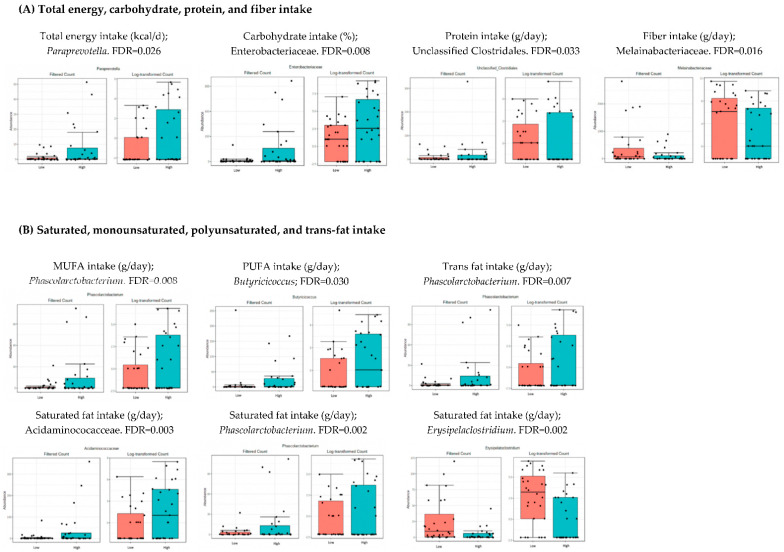
Bacterial taxa differing significantly in relation to total energy, carbohydrate, protein, and fiber intake (**A**), as well as different types of fat: monounsaturated, polyunsaturated, saturated, and trans (**B**) (FDR < 0.05). Red boxes represent participants with lower consumption, and blue boxes represent participants with higher consumption of the specified food group compared to the median.

**Figure 3 nutrients-14-04966-f003:**
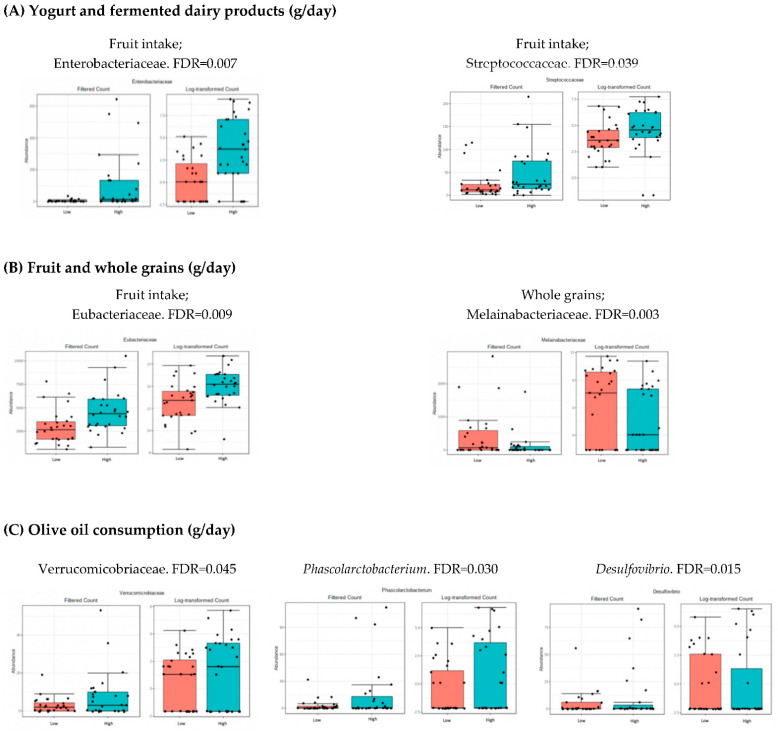
Bacterial taxa differing significantly when comparing individuals with higher and lower consumption of yogurt and fermented dairy (**A**), fruit and whole grains (**B**), and olive oil (**C**) (FDR < 0.05). Red boxes represent participants with lower consumption, and blue boxes represent participants with higher consumption of the specified food group.

**Figure 4 nutrients-14-04966-f004:**
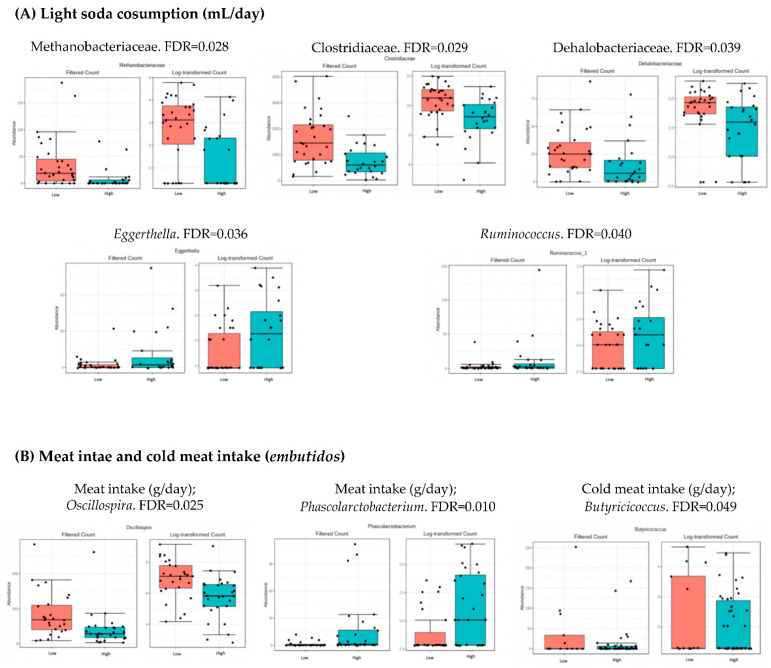
Bacterial taxa differing significantly in relation to the consumption of light soda (**A**) and meat and cold meat intake *(embutidos)* (**B**) (FDR < 0.05). Red boxes represent participants with lower consumption, and blue boxes represent participants with higher consumption of the specified food group compared to the median.

**Table 1 nutrients-14-04966-t001:** Characteristics of the participants.

Variables	All Participants (*n* = 60)	High Group (*n* = 30)	Low Group (*n* = 30)
Age (y)	47.3 ± 11.2	55.8 ± 5.8	38.3 ± 9.3
Weight (kg)	69.9 ± 14.1	79.8 ± 11.6	59.0 ± 5.0
BMI (kg/cm^2^)	24.6 ± 3.9	27.5 ± 2.8	21.4 ± 1.6
Glucose (mg/dL)	94.5 ± 11.3	101.3 ± 11.1	86.3 ± 3.9
Total cholesterol (mg/dL)	219.4 ± 35.4	247.5 ± 21.7	192.8 ± 21.0
HDL (mg/dL)	64.3 ± 15.6	78.0 ± 11.9	52.7 ± 6.8
Triglycerides (mg/dL)	79.0 ± 36.6	102.2 ± 36.1	53.1 ± 9.0
Insulin (µIU/mL)	8.4 ± 4.3	11.4 ± 3.7	5.2 ± 1.6
AST (µ/L)	22.0 ± 14.3	27.3 ± 17.7	16.2 ± 2.4
ALT (µ/L)	22.0 ± 20.6	30.2 ± 25.1	12.8 ± 2.4

Data are expressed as mean ± standard deviation.

**Table 2 nutrients-14-04966-t002:** Nutritional characteristics of the population.

Variables	All (*n* = 60)	High Group (*n* = 30)	Low Group (*n* = 30)	Variables	All (*n* = 60)	High Group (*n* = 30)	Low Group (*n* = 30)
Energy intake (kcal/day)	2381 ± 796	3118 ± 801	1819 ± 299	Yogurt (g/day)	75.4 ± 78.2	129.9 ± 72.4	18.0 ± 21.8
Carbohydrate intake (%)	38.2 ± 7.7	43.8 ± 4.2	31.9 ± 5.3	Fermented dairy (g/day)	93.0 ± 79.8	149.6 ± 72.2	34.3 ± 22.7
Protein intake (%)	18.3 ± 3.4	21.1 ± 3.0	15.9 ± 2.0	Meat (g/day)	169.4 ± 75.6	263.8 ± 166.0	115.1 ± 34.6
Fat intake (%)	41.3 ± 6.8	46.8 ± 5.1	36.3 ± 3.2	Cold meat (g/day)	7.5 ± 9.2	12.8 ± 10.1	1.7 ± 1.7
Fiber intake (g/day)	28.9 ± 11.8	39.9 ± 9.6	20.1 ± 4.7	Olive oil (g/day)	27.6 ± 27.2	45.8 ± 30.5	12.6 ± 5.6
Vegetables (g/day)	432.6 ± 206.8	632.5 ± 194.1	281.3 ± 90.6	Soda (g/day)	13.2 ± 29.5	26.5 ± 36.5	0.0 ± 0.0
Fruit (g/day)	301.8 ± 204.7	527.6 ± 343.2	156.6 ± 65.3	Soda light (g/day)	20.3 ± 42.2	45.9 ± 59.2	0.0 ± 0.0
Legumes (g/day)	23.8 ± 12.6	32.2 ± 12.0	14.9 ± 4.7	Total cholesterol (mg/day)	526.5 ± 210.7	727.4 ± 237.0	379.5 ± 88.0
Cereals (g/day)	166.3 ± 98.5	241.0 ± 83.4	97.1 ± 36.9	Trans fat (g/day)	0.8 ± 0.5	1.1 ± 0.4	0.4 ± 0.1
Whole grains (g/day)	35.8 ± 39.9	64.8 ± 36.8	5.8 ± 8.5	Monounsaturated fat (g/day)	48.8 ± 22.0	69.2 ± 22.7	33.4 ± 7.6
Dairy intake (g/day)	301.3 ± 187.5	442.3 ± 142.3	154.8 ± 80.1	Polyunsaturated fat (g/day)	16.8 ± 6.5	24.3 ± 7.6	11.6 ± 2.7
				Saturated fat (g/day)	30.4 ± 11.9	41.3 ± 8.7	21.4 ± 5.8

Data are expressed as mean ± standard deviation.

## Data Availability

The data presented in this study are available on request from the corresponding author. The data are not publicly available due to privacy restrictions.

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
