# Peer review of "Association of the Gut Microbiota with the Host’s Health through an Analysis of Biochemical Markers, Dietary Estimation, and Microbial Composition"

_nutrients, 2022, doi:10.3390/nu14234966_

Round 1

Reviewer 1 Report

Data collected from 60 adults through food frequency questionnaire, blood plasma/serum measurements, and gut microbiota characterization using 16S rRNA gene sequencing were analyzed and presented to define potential links between food/nutrient consumption, gut microbiota, blood biochemical markers and human health. Authors may wish to make revisions addressing the following issues: 

1. Values of means with standard errors presented in Tables 1 and 2 are good. However, for each of the 33 parameters authors should break down into high and low groups and provide the mean and SD values for the high and low groups respectively, to be consistent with the high and low groups shown in the figures.

2. Current data presentation in the four figures and three supplemental Tables are confusing. Gut microbiota diversity could be classified at various levels: phylum, class, order, family, genus and species. Of the 33 parameters (as listed in the two Tables) and 6 taxonomic levels, gut microbiota data could be used for 198 different group comparisons. Were data presented in the Figures and supplemental Tables represent significant differences found from all these 198 comparisons? Authors should specify at which taxonomic levels gut microbiota data were subjected for comparisons. 

3. Using box plots to present microbiota data is fine in general. It is, however, unnecessary to present both the raw and transformed data. P values, if statistical significant, should be provided above the box plots for readers to clearly see the difference. Also other  forms of data presentation (such as 2D or 3D dot plots, heatmaps etc) should be considered to enhance the appearance of the paper. 

4. The three supplemental tables are all descriptive statements with arrows showing increases or decreases not supported by any statistical analysis. Real data displaying microbiota differences at family and genus levels (using dot plots, heatmaps, box plots etc) should be presented for readers to make their own judgement.

5. Current conclusion paragraph is too long. Authors should condense conclusion to one or two sentences to truly highlight any important (novel) finding from current study.

Author Response

We found the Repetition Rate of your paper is a little high. Please revise the manuscript according to the reviewers' comments. And we found that your paper's repetition rate is 36%. Please revise your paper and make sure that the repetition rate is less than 30%, and then upload the revised file during your revision. You can find the iThenticate report in the attachment.

We have tried to reduce the repetition rate. However, it is not possible in a part of the Material and Methods as we partially describe a technique that has been previously used in other articles. This cannot be considered “repetition”.

However, in the References section, your paper's self-citation of Nutrients is still 17%. The Rate of Self-citations = The number of references from Nutrients / the number of all references.  That is 6 Nutrients papers cited/ 35 Total References = 17%. Kindly decrease the number of Nutrients papers cited or increase the number of Total References to ensure that the self-citation rate is less than 15%.

We have reduced the number of references from Nutrients and substituted them by references from other journals.

If one of the referees has suggested that your manuscript should undergo extensive English revisions, please address this issue during revision. We propose that you use one of the editing services listed at https://www.mdpi.com/authors/english or have your manuscript checked by a native English-speaking colleague.

English has been revised.

We would like to thank the reviewers for their thoughtful comments and suggestions for improving this manuscript. We have considered the comments and tasks proposed by the reviewers and we believe that in addressing these comments, this reviewed manuscript is considerably improved. A detailed point-by-point response to the reviewer’s comments can be found after each question. Changes in the manuscript have been marked up using the “Track Changes” function from MS Word, so any changes can be easily viewed by the editors and reviewers.

Reviewer 1:

  1. Values of means with standard errors presented in Tables 1 and 2 are good. However, for each of the 33 parameters authors should break down into high and low groups and provide the mean and SD values for the high and low groups respectively, to be consistent with the high and low groups shown in the figures.

As suggested by the reviewer, the 33 parameters have been broken down into high and low groups and we now provide also the mean and SD values for the high and low groups in tables 1 and 2.

  1. Current data presentation in the four figures and three supplemental Tables are confusing. Gut microbiota diversity could be classified at various levels: phylum, class, order, family, genus and species. Of the 33 parameters (as listed in the two Tables) and 6 taxonomic levels, gut microbiota data could be used for 198 different group comparisons. Were data presented in the Figures and supplemental Tables represent significant differences found from all these 198 comparisons? Authors should specify at which taxonomic levels gut microbiota data were subjected for comparisons.

As the reviewer indicates, we have performed statistical analyses separately at the different taxonomical levels (from phylum to species). After the statistical analyses at the different taxonomic levels, we have shown only those results that were more relevant because statistically significant after applying the false discovery rate (FDR < 0.05). Except for Verrucomicrobia, we did not observe statistical differences at the levels of phylum, class, order and species. This has been now indicated at the beginning of points 3.3 and 3.4 (Results section).

  1. Using box plots to present microbiota data is fine in general. It is, however, unnecessary to present both the raw and transformed data. P values, if statistical significant, should be provided above the box plots for readers to clearly see the difference. Also other forms of data presentation (such as 2D or 3D dot plots, heatmaps etc) should be considered to enhance the appearance of the paper. 

Thank you. P values (as FDR) have been provided for each box plot. Due to previous experience in the topic, we prefer to maintain the raw and transformed data because it could be of interest for some of the readers, especially for easily comparing our results with their own data.

  1. The three supplemental tables are all descriptive statements with arrows showing increases or decreases not supported by any statistical analysis. Real data displaying microbiota differences at family and genus levels (using dot plots, heatmaps, box plots etc) should be presented for readers to make their own judgement.

Supplementary material represents the same results that have been presented in the figures of the article. As suggested in the comment 2, the statistical significances have been added in the figures. In fact, supplementary material intends to show the results in a more synthesized manner than in the figures of the article. For this reason, these results cannot be shown as figures (it would be duplication). In any case, if the reviewer considers that the supplementary material does not add anything of value to the article, we can eliminate it.

  1. Current conclusion paragraph is too long. Authors should condense conclusion to one or two sentences to truly highlight any important (novel) finding from current study.

Thank you for your comment. Although we think that the reader is now losing important information, the conclusion was shortened according to the suggestion, as follows: This study suggest associations between gut microbiota composition, hepatic health, and insulin resistance status, where the consumption of specific foods related with inflammation features play an important modulatory role.

Reviewer 2

General comments:

Please prepare the article in accordance with the instructions for authors.

Please insert page numbering (as per template) and line numbers:

Page numbering has been added. Line numbers have not been added because they are not in the template.

Please complete the "Citation" details on the front page:

Citation details have been added on the front page.

In the References (not Bibliography) section use Abbreviation name journal and volume number in italic, year in bold, full page range, for a page range use a long "-" from the function insert the symbol instead of the short "-" from the keyboard.

The references have now been formatted according to the instructions.

Detailed comments:

Page 1

Abstract: lack of dependence description gut microbiota composition with HDL-c and insulin levels.

This information has been added to the new version of the Abstract. As it must be shorter than 200 words, we have erased the Introduction sentence and reduced the Material and Methods. 

Does the consumption of olive oil only significantly affect the abundance of Verrucomicobiaceae?

As shown in figure 3C and the Results section, olive oil consumption was related with higher levels of Verrucomicobiaceae and Phascolarctobacterium, but lower levels of Desulfovibrio.

Page 2

Subchapter 2.1, line 6 delete an unnecessary „a dot.”

Corrected

Page 3

What devices were used to register BW and Hight? ?

The devices have been now mentioned in the Material and Methods section. They are the SECA 216 stadiometer (SECA gmnh &co. kg, Hamburg, Germany) and the TANITA SC-330 body composition monitor (Tanita Corp, Tokyo, Japan).

Section 2.4 Please use "⁰" from the insert function at "C":

Corrected

Page 7

Line 3 from the bottom  - Phascolarctobacterium also? Where is the figure?

A sentence about Phascolarctobacterium has been added.

Page 8

Where is Streptoccaceae?, Where are the results of the consumption of yogurt and fermented dairy products on the composition of the gut microbiota, where is the figure for Desulfovibrio in relation to the consumption of olive oil?

The figure for olive oil and Desulfovibrio was in the figure 3C. The other two figures (Streptococcaceae in fruit and Lactobacillaceae in yogurt) had been lost during the formatting of the manuscript. They are now correctly added in the new version of the manuscript.

 Page 12

„References” instead of „Bibliography”:

It has been changed as suggested.

Reviewer 3

The manuscript is interesting and sounding, well written and without ethical issues.

The study methods and results are clearly reported, and data analysis seems to have no biases.

However, the Tab. 1 shows different data in respect of the text. In particular, the age range and mean value, as well as BMI values requires attention.

I suggest revising the whole text to eliminate any possible error in data reporting.

Thank you very much.

We apologize for this mistake. It has been corrected and the whole text has been revised.

Reviewer 2 Report

The aim of this study were to define the association of the gut microbiota composition with the host's helth throught and analysis of biochemical markers. The number of samples used in the experiment is sufficient. The Materials and Methods chapter are written correctly. The Results chapter requires additions and corrections. The discussion is well conducted and comprehensive. Well-chosen references, but must be made according the instructions for authors. Before publishing in Nutrients journal, the article requires additions and corrections. The proposed changes are listed below:

General comments:

Please prepare the article in accordance with the instructions for authors.

Please insert page numbering (as per template) and line numbers

Please complete the "Citation" details on the front page

In the References (not Bibliography) section use Abbreviation name journal and volume number in italic, year in bold, full page range, for a page range use a long "-" from the function insert the symbol instead of the short "-" from the keyboard

Detailed comments:

Page 1

Abstract: lack of dependence description gut microbiota composition with HDL-c and insulin levels.

Does the consumption of olive oil only significantly affect the abundance of Verrucomicobiaceae?

Page 2

Subchapter 2.1, line 6 delete an unnecessary „a dot.”

Page 3

What devices were used to register BW and Hight?

Section 2.4 Please use "⁰" from the insert function at "C"

Page 7

Line 3 from the bottom  - Phascolarctobacterium also? Where is the figure?

Page 8

Where is Streptoccaceae?, Where are the results of the consumption of yogurt and fermented dairy products on the composition of the gut microbiota, where is the figure for Desulfovibrio in relation to the consumption of olive oil?

Page 12

„References” instead of „Bibliography”

Author Response

(The authors gave the same response as above.)

Reviewer 3 Report

The manuscript is interesting and sounding, well written and without ethical issues.

The study methods and results are clearly reported, and data analysis seems to have no biases.

However, the Tab. 1 shows different data in respect of the text. In particular, the age range and mean value, as well as BMI values requires attention.

I suggest to revise the whole text to eliminate any possible error in data reporting.

Author Response

(The authors gave the same response as above.)
